# Reconciling scaling of the optical conductivity of cuprate superconductors with Planckian resistivity and specific heat

Bastien Michon [1,2,3], Christophe Berthod [1], Carl Willem Rischau[1], Amirreza Ataei [4], Lu Chen [4], Seiki Komiya[5], Shimpei Ono [5], Louis Taillefer [4,6], Dirk van der Marel [1] ✉ & Antoine Georges [1,7,8,9] ✉

Materials tuned to a quantum critical point display universal scaling properties as a function of temperature $T$ and frequency $\omega$. A long-standing puzzle regarding cuprate superconductors has been the observed power-law dependence of optical conductivity with an exponent smaller than one, in contrast to $T$-linear dependence of the resistivity and $\omega$-linear dependence of the optical scattering rate. Here, we present and analyze resistivity and optical conductivity of $La_{2-x}Sr_xCuO_4$ with $x = 0.24$. We demonstrate $\hbar\omega/k_BT$ scaling of the optical data over a wide range of frequency and temperature, $T$-linear resistivity, and optical effective mass proportional to $\sim \ln T$ corroborating previous specific heat experiments. We show that a $T, \omega$-linear scaling *Ansatz* for the inelastic scattering rate leads to a unified theoretical description of the experimental data, including the power-law of the optical conductivity. This theoretical framework provides new opportunities for describing the unique properties of quantum critical matter.

The linear-in-temperature electrical resistivity is one of the remarkable properties of the cuprate high temperature superconductors[1–4]. By means of chemical doping, it is possible to tune these materials to a carrier concentration where $\rho(T) = \rho_0 + AT$ in a broad temperature range. For $Bi_{2+x}Sr_{2-y}CuO_{6±\delta}$, it has been possible to demonstrate this from 7 to 700 K[5] by virtue of the low $T_c$ of this material. For the underdoped cuprates, the linear-in-$T$ resistivity is ubiquitous for temperatures $T > T^*$, where $T^*$ is a doping-dependent cross-over temperature that decreases as a function of doping and vanishes at a critical doping $p^*$. From one cuprate family to another, the exact value of $p^*$ varies widely within the range $0.19 < p^* < 0.40$[6–10]. For doping levels $p < p^*$, many of the physical properties indicate the presence of a pseudogap that vanishes at $p^*$[11,12]. When $p$ is tuned exactly to $p^*$, the $T$-linear resistivity persists down to $T = 0$ K if superconductivity is

suppressed e.g. by applying a magnetic field[13–15]. The conundrum of the $T$-linear resistivity has been associated to the idea that the momentum relaxation rate cannot exceed the Planckian dissipation $k_BT/\hbar$[16–19], a state of affairs for which there exists now strong experimental support[20,21].

As expected for a system tuned to a quantum critical point[22], $\hbar\omega/k_BT$ scaling has been observed in the optical properties of high-$T_c$ cuprates[23,24] over some range of doping. The optical scattering rate obtained from an extended Drude fit to the data was found to obey a $T$-linear dependence $1/\tau \sim k_BT/\hbar$ in the low-frequency regime ($\hbar\omega \lesssim 1.5k_BT$) as well as a linear dependence on energy over an extended frequency range[24–29]. A direct measurement of the linear temperature dependence of the single-particle relaxation rate extending over 70% of the Fermi surface was obtained with angle resolved

[1]Department of Quantum Matter Physics, University of Geneva, Geneva, Switzerland. [2]Department of Physics, City University of Hong Kong, Kowloon, Hong Kong, China. [3]Hong Kong Institute for Advanced Study, City University of Hong Kong, Kowloon, Hong Kong, China. [4]Institut Quantique, Département de Physique & RQMP, Université de Sherbrooke, Sherbrooke, Québec, Canada. [5]Energy Transformation Research Laboratory, Central Research Institute of Electric Power Industry, Yokosuka, Kanagawa, Japan. [6]Canadian Institute for Advanced Research, Toronto, ON, Canada. [7]Collège de France, Paris, France. [8]Center for Computational Quantum Physics, Flatiron Institute, New York, NY, USA. [9]CPHT, CNRS, École Polytechnique, IP Paris, Palaiseau, France. ✉e-mail: dirk.vandermarel@unige.ch; antoine.georges@college-de-france.fr

photoemission spectroscopy (ARPES)[30]. These observations are qualitatively consistent with the $T$-linear dependence of the resistivity and Planckian behavior. In contrast, by analyzing the modulus and phase of the optical conductivity itself, a power-law behavior $\sigma(\omega) = C/(-i\omega)^{\nu^*}$ with an exponent $\nu^* < 1$ was reported at higher frequencies $\hbar\omega \gtrsim 1.5 k_B T$[23,24,28,29,31,32]. The exponent was found to be in the range $\nu^* \approx 0.65$ with some dependence on sample and doping level[23,26,28,29]. Hence, from these previous analyses, it would appear that different power laws are needed to describe optical spectroscopy data: one at low frequency consistent with $\hbar\omega/k_B T$ scaling and Planckian behavior ($\nu = 1$) and another one with $\nu^* < 1$ at higher frequency, most apparent on the optical conductivity itself in contrast to $1/\tau$. A number of theoretical approaches have considered a power-law dependence of the conductivity[33–42] without resolving this puzzle. A notable exception is the work of Norman and Chubukov[43]. The basic assumption of this work is that the electrons are coupled to a Marginal Fermi Liquid susceptibility[3,4,44,45]. The logarithmic behavior of the susceptibility and corresponding high-energy cut-off observed to be ~ 0.4 eV with ARPES[46], is responsible for the apparent sub-linear power law behavior of the optical conductivity. Our work broadens and amplifies this observation. A quantitative description of all aspects at low and high energy in one fell swoop has, to the best of our knowledge, not been presented to this day.

Here we present systematic measurements of the optical spectra, as well as dc resistivity, of a La$_{2-x}$Sr$_x$CuO$_4$ (LSCO) sample with $x = p = 0.24$ close to the pseudogap critical point, over a broad range of temperature and frequency. We demonstrate that the data display Planckian quantum critical scaling over an unprecedented range of $\hbar\omega/k_B T$. Furthermore, a direct analysis of the data reveals a logarithmic temperature dependence of the optical effective mass. This

establishes a direct connection to another hallmark of Planckian behavior, namely the logarithmic enhancement of the specific heat coefficient $C/T \sim \ln T$ previously observed for LSCO at $p = 0.24$[47] as well as for other cuprate superconductors such as Eu-LSCO and Nd-LSCO[48].

We introduce a theoretical framework which relies on a minimal Planckian scaling *Ansatz* for the inelastic scattering rate. We show that this provides an excellent description of the experimental data. Our theoretical analysis offers, notably, a solution to the puzzle mentioned above. Indeed we show that, despite the purely Planckian *Ansatz* which underlies our model, the optical conductivity computed in this framework is well described by an apparent power law with $\nu^* < 1$ over an intermediate frequency regime, as also observed in our experimental data. The effective exponent $\nu^*$ is found to be non-universal and to depend on the inelastic coupling constant, which we determine from several independent considerations. The proposed theoretical analysis provides a unifying framework in which the behavior of the $T$-linear resistivity, $\ln T$ behavior of $C/T$, and scaling properties of the optical spectra can all be understood in a consistent manner.

## Results
### Optical spectra and resistivity
We measured the optical properties and extracted the complex optical conductivity $\sigma(\omega, T)$ of an LSCO single crystal with a-b orientation (CuO$_2$ planes). The hole doping is $p = x = 0.24$, which places our sample above and close to the pseudogap critical point of the LSCO family[7,14,49]. The pseudogap state for $T < T^*$, $p < p^*$ is well characterized by transport measurements[12] and ARPES[11]. The relatively low $T_c = 19$ K of this sample is interesting for extracting the normal-state properties in optics down to low temperatures without using any external magnetic field. In particular, this sample is the same LSCO $p = 0.24$ sample as in Ref. 50, where the evolution of optical spectral weights as a function of doping was reported.

The quantity probed by the optical experiments of the present study is the planar complex dielectric function $\epsilon(\omega)$. The dielectric function has contributions from the free charge carriers, as well as interband (bound charge) contributions. In the limit $\omega \to 0$, the latter contribution converges to a constant real value, traditionally indicated with the symbol $\epsilon_\infty$:

$$\epsilon(\omega) = \epsilon_\infty + i\frac{\sigma(\omega)}{\epsilon_0 \omega} \tag{1}$$

$$\sigma(\omega) = i\frac{e^2 K/(\hbar d_c)}{\hbar\omega + M(\omega)}. \tag{2}$$

Here the free-carrier response $\sigma(\omega)$ is given by the generalized Drude formula, where all dynamical mass renormalization ($m^*/m$) and relaxation ($\hbar/\tau$) processes are represented by a memory-function[51,52]

$$M(\omega) = \hbar\omega\left[\frac{m^*(\omega)}{m} - 1\right] + i\frac{\hbar}{\tau(\omega)}. \tag{3}$$

The free-carrier spectral weight per plane is given by the constant $K$ and the interplanar spacing is $d_c$. The scattering rate $\hbar/\tau(\omega)$ deduced using Eqs. ((1), (2), (3)) and the values of $K$ and $\epsilon_\infty$ discussed below are displayed in Fig. 1c. It depends linearly on frequency for $k_B T \ll \hbar\omega \lesssim 0.4$ eV and approaches a constant value for $\hbar\omega < k_B T$. This behavior is similar to that reported for Bi2212[23]. The sign of the curvature above 0.4 eV depends on $\epsilon_\infty$ and changes from positive to negative near $\epsilon_\infty = 4.5$. Our determination $\epsilon_\infty = 2.76$ presented in Scaling analysis does not take into account data for $\hbar\omega > 0.4$ eV and may therefore yield unreliable values of $\hbar/\tau$ in that range (see Supplementary Information Sec. A and B).

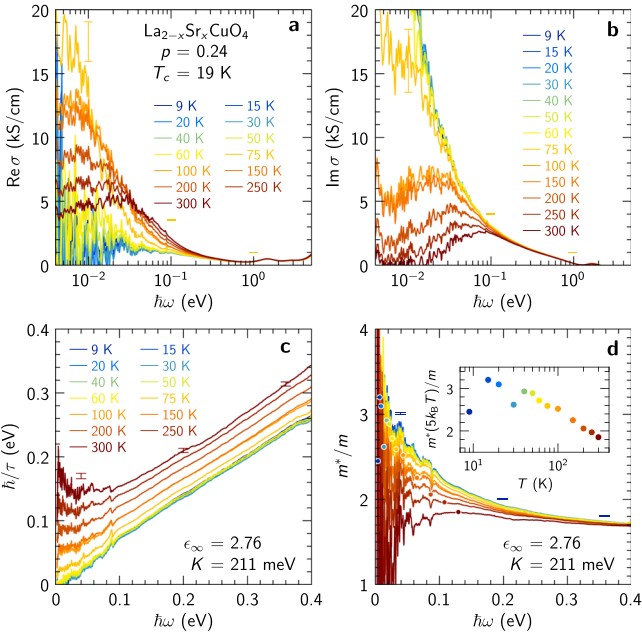

**Fig. 1 | Optical data of La$_{2-x}$Sr$_x$CuO$_4$ at $p = 0.24$. a** Real and **b** imaginary part of the optical conductivity $\sigma$ deduced from the dielectric function $\epsilon$ (Supplementary Fig. 1), using Eq. (14) and the value $\epsilon_\infty = 2.76$. **c** Scattering rate and **d** effective mass deduced from Eqs. (16) and (17) using $K = 211$ meV. The values of $\epsilon_\infty$ and $K$ are discussed and justified in the text. Inset: Temperature dependence of $m^*/m$ at $\hbar\omega = 5k_B T$ (see dots in **d**). In each panel errorbars are indicated for three representative frequencies and pertain to the upper curve, *i.e.*, the lowest temperature for $\sigma(\omega)$, $m^*(\omega)/m$ and the highest temperature for $\hbar/\tau(\omega)$. They represent the uncertainty arising from reflectivity calibration using in-situ gold evaporation, and have been estimated by repeating the Kramers–Kronig analysis after multiplying the reflectivity curves by $1 \pm 0.002$.

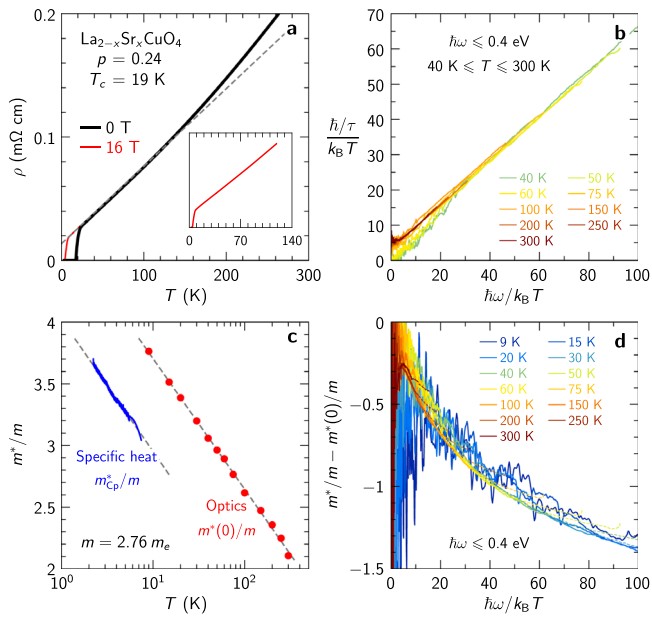

**Fig. 2 | Scaling of scattering rate and mass enhancement. a** Temperature-dependent resistivity measured in zero field (black) and at 16 teslas (red). The inset emphasizes the linearity of the 16 T data at low temperature. The dashed line shows $\rho_0 + AT$ with $\rho_0 = 12.2\,\mu\Omega$cm and $A = 0.63\,\mu\Omega$cm/K. **b** Scattering rate divided by temperature plotted versus $\omega/T$; the collapse of the curves indicates a behavior $1/\tau \sim Tf_\tau(\omega/T)$. **c** Effective quasiparticle mass (in units of the indicated band mass $m$) deduced from the low-temperature electronic specific heat[47] $[m^*_{Cp} = (3/\pi)(\hbar^2 d_c/k_B^2)(C/T)]$ and zero-frequency optical mass enhancement; the dashed lines indicate $\ln T$ behavior. **d** Optical mass minus the zero-frequency mass shown in **c** plotted versus $\omega/T$; the collapse of the curves indicates a behavior $m^*(\omega) - m^*(0) \sim f_m(\omega/T)$. The data between 0.22 and 0.4 eV are shown as dotted lines. $\epsilon_\infty = 2.76$ was used here as in Fig. 1.

This linear dependence of the scattering rate calls for a comparison with resistivity. Hence we have also measured the temperature dependence of the resistivity of our sample under two magnetic fields $H = 0$ T and $H = 16$ T. As displayed in Fig. 2a, the resistivity has a linear $T$-dependence $\rho = \rho_0 + AT$ over an extended range of temperature, with $A \approx 0.63\,\mu\Omega$cm/K. This is a hallmark of cuprates in this regime of doping[10,13,14,20,53]. It is qualitatively consistent with the observed linear frequency dependence of the scattering rate and, as discussed later in this paper, also in good quantitative agreement with the $\omega \to 0$ extrapolation of our optical data within experimental uncertainties.

The optical mass enhancement $m^*(\omega)/m$ is displayed in Fig. 1d. With the chosen normalization, $m^*/m$ does not reach the asymptotic value of one in the range $\hbar\omega < 0.4$ eV, which means that intra- and interband and/or mid-infrared transitions overlap above 0.4 eV. The inset of Fig. 1d shows a semi-log plot of the mass enhancement evaluated at $\hbar\omega = 5k_BT$, where the noise level is low for $T \geqslant 40$ K. Despite the larger uncertainties at low $T$, this plot clearly reveals a logarithmic temperature dependence of $m^*/m$. This is a robust feature of the data, independent of the choice of $\epsilon_\infty$ and $K$. We note that the specific heat coefficient $C/T$ of LSCO at the same doping level was previously reported to display a logarithmic dependence on temperature, see Fig. 2c[47,48]. We will further elaborate on this important finding of a logarithmic dependence of the optical mass and discuss its relation to specific heat in the next section.

**Scaling analysis**

In this section, we consider simultaneously the frequency and temperature dependence of the optical properties and investigate whether $\hbar\omega/k_BT$ scaling holds for this sample close to the pseudogap critical

point. We propose a procedure to determine the three parameters $\epsilon_\infty$, $K$, and $m$ introduced above.

**Putting $\omega/T$ scaling to the test.** Quantum systems close to a quantum critical point display scale invariance. Temperature being the only relevant energy scale in the quantum critical regime, this leads in many cases to $\omega/T$ scaling[22] (in most of the discussion below, we set $\hbar = k_B = 1$ except when mentioned explicitly). In such a system we expect the complex optical conductivity to obey a scaling behavior $1/\sigma(\omega,T) \propto T^\nu F(\omega/T)$, with $\nu \leqslant 1$ a critical exponent. More precisely, the scaling properties of the optical scattering rate and effective mass read:

$$1/\tau(\omega,T) = T^\nu f_\tau(\omega/T) \qquad (4)$$

$$m^*(\omega,T) - m^*(0,T) = T^{\nu-1}f_m(\omega/T) \qquad (5)$$

with $f_\tau$ and $f_m$ two scaling functions. This behavior requires that both $\hbar\omega$ and $k_BT$ are smaller than a high-energy electronic cutoff, but their ratio can be arbitrary. Furthermore, we note that when $\nu = 1$ (Planckian case) the scaling is violated by logarithmic terms, which control in particular the zero-frequency value of the optical mass $m^*(0,T)$. As shown in Theory within a simple theoretical model, $\omega/T$ scaling nonetheless holds in this case to an excellent approximation provided that $m^*(0,T)$ is subtracted, as in Eq. (5). We also note that in a Fermi liquid, the single-particle scattering rate $\propto \omega^2 + (\pi T)^2$ does obey $\omega/T$ scaling (with formally $\nu = 2$), but the optical conductivity does not. Indeed, it involves $\omega/T^2$ terms violating scaling, and hence depends on two scaling variables $\omega/T^2$ and $\omega/T$, as is already clear from an (approximate) generalized Drude expression $1/\sigma \sim -i\omega + \tau_0[\omega^2 + (2\pi T)^2]$. For a detailed discussion of this point, see Ref. 54. Such violations of scaling by $\omega/T^\nu$ terms apply more generally to the case where the scattering rate varies as $T^\nu$ with $\nu > 1$. Hence, $\omega/T$ scaling for both the optical scattering rate and optical effective mass are a hallmark of non-Fermi liquid behavior with $\nu \leqslant 1$. Previous work has indeed provided evidence for $\omega/T$ scaling in the optical properties of cuprates[23,24].

Here, we investigate whether our optical data obey $\omega/T$ scaling. We find that the quality of the scaling depends sensitively on the chosen value of $\epsilon_\infty$. Different prescriptions in the literature to fix $\epsilon_\infty$ yield—independently of the method used—values ranging from $\epsilon_\infty \approx 4.3$ for strongly underdoped Bi2212 to $\epsilon_\infty \approx 5.6$ for strongly overdoped Bi2212[32,55]. The parameter $\epsilon_\infty$ is commonly understood to represent the dielectric constant of the material in the absence of the charge carriers, and is caused by the bound charge responsible for interband transitions at energies typically above 1 eV. While this definition is unambiguous for the insulating parent compound, for the doped material one is confronted with the difficulty that the optical conductivity at these higher energies also contains contributions described by the self-energy of the conduction electrons, caused for example by their coupling to dd-excitations[56]. Consequently, not all of the oscillator strength in the interband region represents bound charge. Our model overcomes this hurdle by determining the low-energy spectrum below 0.4 eV, and subsuming all bound charge contributions in a single constant $\epsilon_\infty$. Its value is expected to be bound from above by the value of the insulating phase, in other words we expect to find $\epsilon_\infty < 4.5$ (see Supplementary Information Sec. A). Rather than setting an a priori value for $\epsilon_\infty$, we follow here a different route and we choose the value that yields the best scaling collapse for a given value of the exponent $\nu$. This program is straightforwardly implemented for $1/\tau$ and indicates that the best scaling collapse is achieved with $\nu \approx 1$ and $\epsilon_\infty \approx 3$, see Fig. 2b as well as Supplementary Information Sec. B and Supplementary Fig. 2. Turning to $m^*$, we found that subtracting the dc value $m^*(\omega = 0, T)$ is crucial when attempting to collapse the data. Extrapolating optical data to zero frequency is hampered by noise. Hence,

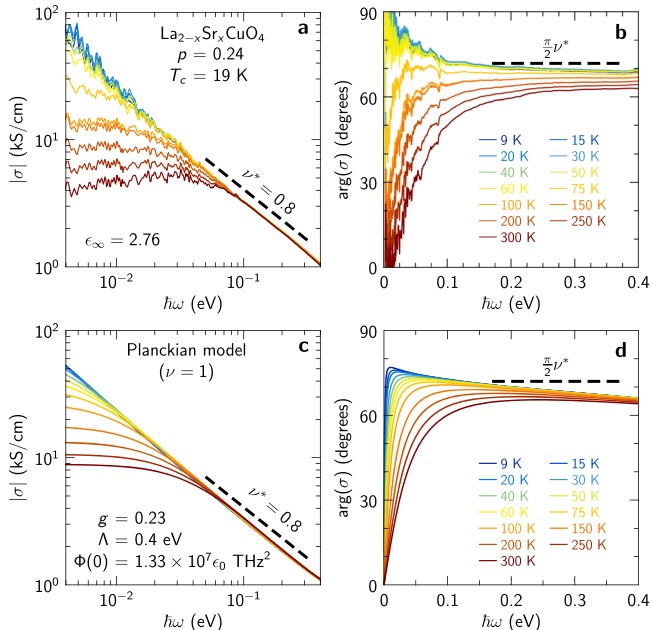

**Fig. 3 | Sub-linear power law at intermediate frequencies. a** Modulus and **b** phase of the complex conductivity shown in Fig. 1a and b; the modulus decays with an exponent $\nu^* \approx 0.8$ and the phase approaches a value slightly lower than $(\pi/2)\nu^*$. **c** and **d**: same quantities calculated using a Planckian model with linear-in-energy scattering rate, Eqs. (7) and (10). The model and parameters are discussed in the text.

instead of attempting an extrapolation, we consider $m^*(0, T)$ as adjustable values that we again tune such as to optimize the collapse of the optical data. This analysis of $m^*/m$ confirms that the best scaling collapse occurs for $\nu \approx 1$ but indicates a larger $\epsilon_\infty \approx 7$ (Supplementary Information Sec. B and Supplementary Fig. 3). The determination of $\epsilon_\infty$ from the mass data depends sensitively on the frequency range tested for scaling and drops to value below $\epsilon_\infty = 3$ when focusing on lower frequencies. As a third step, we perform a simultaneous optimization of the data collapse for $1/\tau$ and $m^*/m$, which yields the values $\nu = 1$, $\epsilon_\infty = 2.76$ which we will adopt throughout the following. Note that a determination of $\epsilon_\infty$ by separation of the high-frequency modes in a Drude–Lorentz representation of $\epsilon(\omega)$ yields a larger value $\epsilon_\infty = 4.5 \pm 0.5$, as typically found in the cuprates[23,32,57]. Importantly, all our conclusions hold if we use this latter value in the analysis, however, the quality of the scaling displayed in Figs. 2 and 5 is slightly degraded.

**Scaling of the optical scattering rate and connection to resistivity.** The scaling properties of the scattering rate obtained from our optical data according to the procedure described above is illustrated in Fig. 2b, which displays $\hbar/\tau$ divided by $k_B T$ and plotted versus $\hbar\omega/k_B T$ for temperatures above the superconducting transition. The collapse of the curves at different temperatures reveals the behavior $\hbar/\tau \propto T f_\tau(\omega/T)$. The function $f_\tau(x)$ reaches a constant $f_\tau(0) > 0$ at small values of the argument, and behaves for large arguments as $f_\tau(x \gg 1) \propto x$. This is consistent with the typical quantum critical behavior $\hbar/\tau \sim \max(T, \omega)$. When inserted in the $\omega = 0$ limit of Eq. (15), the value $f_\tau(0) \approx 5$ indicated by Fig. 2b yields $1/\sigma(0) = AT$ with $A = 0.55 \, \mu\Omega$ cm/K, in fairly good agreement with the measured resistivity (Fig. 2a). Hence the resistivity and optical-spectroscopy data are fully consistent, both of them supporting a Planckian dissipation scenario with $\nu = 1$ for LSCO at $p = 0.24$.

**Spectral weight, effective mass and connection to specific heat.** The dc mass enhancement values $m^*(0, T)/m$ resulting from the procedure described above are displayed in Fig. 2c. Remarkably, as seen on this figure, the scaling analysis delivers an almost perfectly

logarithmic temperature dependence of $m^*(0, T)$, consistent with a Planckian behavior $\nu = 1$. As mentioned above, this logarithmic behavior can actually be identified in the unprocessed optical data, (see inset of Fig. 1). In order to compare this behavior to the corresponding logarithmic behavior reported for the specific heat, we note that the scaling analysis provides $m^*(0, T)$ up to a multiplicative constant $Km$, where $m$ is the band mass. In contrast, the electronic specific heat yields the quasiparticle mass in units of the bare electron mass $m_e$. We expect that the logarithmic $T$-variation of $m^*(0, T)$ and $m_{qp}^* \propto C/T$ are both due to the critical inelastic scattering and that the $\ln T$ term in each quantity should therefore have identical prefactors. Imposing this identity provides a relationship between $Km$ and $m_e$, namely $(m/m_e) K = 583$ meV.

Remarkably, we have found that this condition is obeyed within less than a percent by a square-lattice tight-binding model with parameters appropriate for LSCO at $p = 0.24$ (Supplementary Information Sec. E). This model has nearest and next-nearest neighbor hopping amplitudes $t = 0.3$ eV and $t'/t = -0.17$[58], respectively, and an electronic density $n = 0.76/a^2$. The Fermi-level density of states is $1.646/(eVa^2)$, which corresponds to a band mass $m/m_e = 2.76$ using the LSCO lattice parameter $a = 3.78$ Å. The spectral weight is $K = 211$ meV, such that the prediction of this tight-binding model is $(m/m_e)K = 582$ meV, in perfect agreement with the previously determined value. In view of this agreement, we use the tight-binding model in order to fix the remaining two system parameters: $m = 2.76 \, m_e$ and $K = 211$ meV. Figure 2c compares the mass enhancement inferred from the low-temperature specific heat and from the scaling analysis of the optical data. The tight-binding value of the product $Km$ ensures that both data sets have the same slope on a semi-log plot. However, the resulting optical mass enhancement is larger than the quasiparticle mass enhancement by $\approx 0.75$, which is also the amount by which the infrared mass enhancement exceeds unity in Fig. 1d. A mass enhancement larger than unity at 0.4 eV implies that part of the intraband spectral weight lies above 0.4 eV, overlapping with the interband transitions. Conversely, interband spectral weight is likely leaking below 0.4 eV, which prevents us from accessing the absolute value of the genuine intraband mass by optical means. Figure 2d shows the collapse of the frequency-dependent change of the mass enhancement, confirming the behavior $m^*(\omega) - m^*(0) \approx T^{\nu-1} f_m(\omega/T)$ with $\nu = 1$. The shape of the scaling function $f_m(x)$ agrees remarkably well with the theoretical prediction derived in Theory below.

**Apparent power-law behavior: a puzzle.** The above scaling analysis has led us to the following conclusions. (i) The optical scattering rate and optical mass enhancement of LSCO at $p = 0.24$ exhibit $\omega/T$ scaling over two decades for the chosen value $\epsilon_\infty = 2.76$. (ii) The best collapse of the data is achieved for an exponent $\nu = 1$ corresponding to Planckian dissipation. This behavior is consistent with the measured $T$-linear resistivity. (iii) The temperature dependence of $m^*(0, T)$ that produces the best data collapse is logarithmic, consistently with the temperature dependence of the electronic specific heat.

Hence, the data presented in Fig. 2 provide compelling evidence that the low-energy carriers in LSCO at the doping $p = 0.24$ experience linear-in-energy and linear-in-temperature inelastic scattering processes, as expected in a scale-invariant quantum critical system characterized by Planckian dissipation. It is therefore at first sight surprising that the infrared conductivity exhibits as a function of frequency a power law with an exponent that is clearly smaller than unity, as highlighted in Fig. 3a, b. These figures show that the modulus and phase of $\sigma$ are both to a good accuracy consistent with the behavior $\sigma \propto (-i\omega)^{-\nu^*} = \omega^{-\nu^*} e^{i\frac{\pi}{2}\nu^*}$ with an exponent $\nu^* = 0.8$. A similar behavior with exponent $\nu^* \approx 0.6$ was reported for optimally- and overdoped Bi2212[23], while earlier optical investigations of YBCO and Bi2212 have also reported power law behavior of Re $\sigma(\omega)$[26,28,29]. We now address this question by considering a theoretical model presented in the following

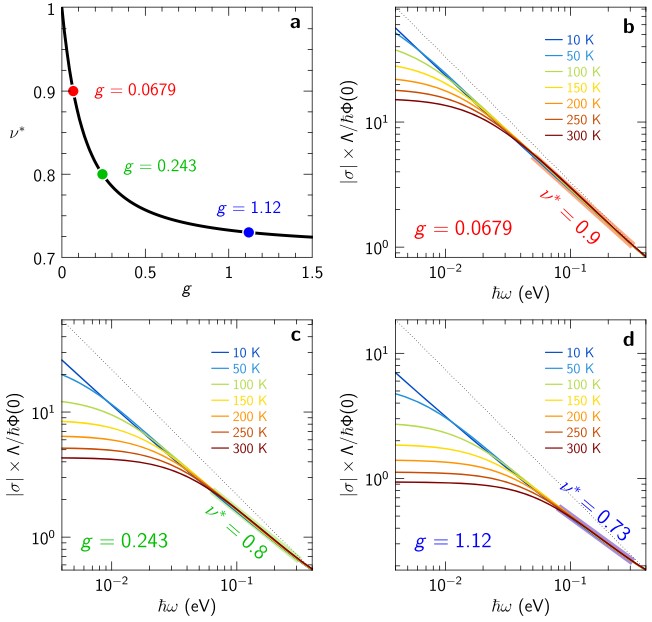

**Fig. 4 | Effective exponent.** Emergence of an apparent sub-linear power-law in a pure Planckian model. **a** Apparent exponent given by Eq. (12) versus interaction strength $g$. **b**–**d** Modulus of the optical conductivity on a log-log scale showing the apparent power law at energies between $k_B T$ and the cutoff $\Lambda = 0.4$ eV. Data are shown for three values of $g$ (dots in **a**) and a range of temperatures. Both horizontal and vertical axes cover exactly two decades, such that a $1/\omega$ behavior would correspond to a slope of $-1$ (dotted line).

section. As derived there, and illustrated in Fig. 3c, d, we show that an apparent exponent $\nu^* < 1$ is actually *predicted by theory* for Planckian systems with single-particle self-energy exponent $\nu = 1$, over an intermediate range of values of $\omega/T$. This is one of the central claims of our work.

## Theory

In this section, we consider a simple theoretical model and explore its implications for the optical conductivity. Our central assumption is that the inelastic scattering rate (imaginary part of the self-energy) obeys the following scaling property:

$$-\mathrm{Im}\,\Sigma(\varepsilon) = g\pi k_B T\, S\!\left(\frac{\varepsilon}{k_B T}\right). \tag{6}$$

In this expression $g$ is a dimensionless inelastic coupling constant and $\varepsilon = \hbar\omega$. This $\hbar\omega/k_B T$ scaling form is assumed to apply when both $\hbar\omega$ and $k_B T$ are smaller than a high-energy cutoff $\Lambda$ but their ratio can be arbitrary. The detailed form of the scaling function $S$ is not essential, except for the requirements that $S(0)$ is finite and $S(x \gg 1) \propto |x|$. These properties ensure that the low-frequency inelastic scattering rate depends linearly on $T$ for $\hbar\omega \ll k_B T$ and that dissipation is linear in energy for $\hbar\omega \gg k_B T$, which are hallmarks of Planckian behavior. We note that such a scaling form appears in the context of microscopic models such as overscreened non-Fermi liquid Kondo models[59] and the doped SYK model close to a quantum critical point[60–64]. In such models, conformal invariance applies and dictates the form of the scaling function to be $S(x) = x \coth(x/2)$ (with possible modifications accounting for a particle-hole spectral asymmetry parameter, see Refs. 59,65 and Supplementary Information Sec. F. We have assumed that the inelastic scattering rate is momentum independent (spatially local) i.e. uniform along the Fermi surface. This assumption is supported by recent angular-dependent magnetoresistance experiments on Nd-LSCO at a doping close to the pseudogap quantum critical point[21]—see

also Ref. 66. In contrast, the elastic part of the scattering rate (not included in our theoretical model) was found to be strongly anisotropic (angular dependent).

The real part of the self-energy is obtained from the Kramers–Kronig relation which reads, substituting the scaling form above:

$$\Sigma(z) = g k_B T \int_\Lambda dx\, \frac{S(x)}{z/k_B T - x}. \tag{7}$$

We note that this expression is only defined provided the integral is bounded at high-frequency by the cutoff $\Lambda$, as detailed in Supplementary Information Sec. C. This reflects into a logarithmic temperature dependence at low energy:

$$\mathrm{Re}\,[\Sigma(\varepsilon) - \Sigma(0)] = -2g\varepsilon \ln(a\Lambda/k_B T) \tag{8}$$

with $a = 0.770542$ a numerical constant (Supplementary Information Sec. C). Correspondingly, the effective mass of quasiparticles, as well as the specific heat, is logarithmically divergent at low temperature:

$$\frac{m^*_{\mathrm{qp}}}{m} = \frac{1}{Z} = 1 + 2g \ln\!\left(a\frac{\Lambda}{k_B T}\right) \tag{9}$$

with $1/Z = 1 - d\mathrm{Re}\,\Sigma(\varepsilon)/d\varepsilon|_{\varepsilon=0}$. Importantly, the coefficient of the dominant $\ln T$ term depends only on the value of the inelastic coupling $g$.

In a local (momentum-independent) theory, vertex corrections are absent[67,68] and the optical conductivity can thus be directly computed from the knowledge of the self-energy as[69]:

$$\sigma(\omega) = \frac{i\Phi(0)}{\omega} \int_{-\infty}^{\infty} d\varepsilon\, \frac{f(\varepsilon) - f(\varepsilon + \hbar\omega)}{\hbar\omega + \Sigma^*(\varepsilon) - \Sigma(\varepsilon + \hbar\omega)} \tag{10}$$

where $f(\varepsilon) = (e^{\varepsilon/k_B T} + 1)^{-1}$ is the Fermi function and $\Sigma^*$ denotes complex conjugation. In this expression $\Phi(\varepsilon) = 2(e/\hbar)^2 \int_{\mathrm{BZ}} \frac{d^2 k}{(2\pi)^2} (\partial \varepsilon_{\mathbf{k}}/\partial k_x)^2 \delta(\varepsilon + \mu_0 - \varepsilon_{\mathbf{k}})$ is the transport function associated with the bare bandstructure. We have assumed that its energy dependence can be neglected so that only the value $\Phi(0)$ at the Fermi level matters (we set $\mu_0 = 0$ by convention). Using a tight-binding model for the band dispersion, $\Phi(0)$ can be related to the spectral weight $K$ discussed in the previous section as: $(\hbar/e)^2 \Phi_{2D}(0) = K = 211$ meV, i.e. $\Phi(0) = \Phi_{2D}(0)/d_c = 1.33 \times 10^7 \epsilon_0 \mathrm{THz}^2$ (see Supplementary Information Sec. E).

Within our model, the behavior of the optical conductivity relies on three parameters: the cutoff $\Lambda$, the Drude weight related to $\Phi(0)$ and, importantly, the dimensionless inelastic coupling $g$. An analysis of Eq. (10), detailed in Supplementary Information Sec. C, yields the following behavior in the different frequency regimes:

- $\hbar\omega \lesssim k_B T$. The optical conductivity in this regime takes a Drude-like form Eq. (15) with $\hbar/\tau = 4\pi g k_B T$. The numerically computed zero-frequency optical mass enhancement $m^*(0)/m$ agrees very well with $m^*_{\mathrm{qp}}/m = 1/Z$ as given by Eq. (9), see Supplementary Fig. 6. Fitting Eq. (9) to the $m^*(0)/m$ data in Fig. 2c provides the values $g = 0.23$ and $\Lambda = 0.4$ eV.

- $\hbar\omega \gtrsim \Lambda$. In this high-frequency regime, the asymptotic behavior is fixed by causality and reads $|\sigma| \cdot 1/\omega$, $\arg(\sigma) \to \pi/2$ (see Supplementary Fig. 4 and Supplementary Fig. 5).

- $k_B T \lesssim \hbar\omega \lesssim \Lambda$. In this regime, which is the most important in practice when considering our experimental data, one can derive the following expression:

$$\sigma(\omega) \approx \frac{\Phi(0)}{-i\omega} \frac{1}{1 + 2g\left[1 - \ln\!\left(\frac{\hbar\omega}{2\Lambda}\right)\right] + i\pi g}. \tag{11}$$

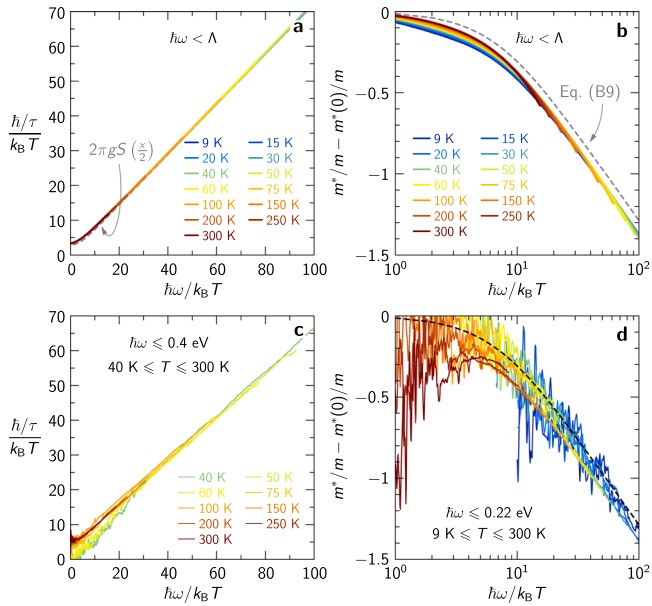

**Fig. 5 | Frequency-temperature scaling. a** Approximate collapse of the theoretical scattering rate and **b** mass enhancement; the dashed lines show $2\pi g S(x/2)$ in **a** and Eq. (S16) in **b**. **c** Same data as in Fig. 2b. **d** Same data as in Fig. 2d on a logarithmic scale (not displayed here because of excessive noise: $\hbar\omega/k_B T < 10$ for $T < T_c$); the dashed line is Eq. (S16).

Remarkably, as shown in Fig. 4, the theoretical optical conductivity is very well approximated in this regime by an apparent power-law dependence $|\sigma| \sim |\omega|^{-\nu^*}$, over at least a decade in frequency. The effective exponent $\nu^* < 1$ depends continuously on the inelastic coupling constant $g$ and can be estimated as:

$$\nu^* \equiv -\frac{d \ln |\sigma|}{d \ln \omega}\Big|_{\hbar\omega = \Lambda/2}$$
$$\equiv 1 - \frac{2g[1 + 2g(1 + \ln 4)]}{\pi^2 g^2 + [1 + 2g(1 + \ln 4)]^2}. \quad (12)$$

Correspondingly, $\arg(\sigma)$ has a plateau at $\arg(\sigma) \approx \pi\nu^*/2$ before reaching its eventual asymptotic value $\pi/2$ (Supplementary Fig. 5). Using the value $g = 0.23$ deduced above from $m^*(0)/m$ yields $\nu^* = 0.8$, in very good agreement with experiment, as shown in Fig. 3.

In the dc limit $\omega \to 0$, Eq. (10) together with our *Ansatz* for the scattering rate, yields a $T$-linear resistivity:

$$\rho = AT, \quad A = \frac{4\pi^3 k_B}{7\zeta(3)\hbar} \frac{g}{\Phi(0)} = \frac{4\pi^3 \hbar k_B d_c}{7\zeta(3)e^2} \frac{g}{K}. \quad (13)$$

Using the values of $g$ and $\Phi(0)$ determined above, we obtain: $A = 0.38\ \mu\Omega\text{cm/K}$, to be compared to the experimental value $A = 0.63\ \mu\Omega\text{cm/K}$. It is reassuring that a reasonable order of magnitude is obtained (at the 60% level) for the $A$-coefficient, while obviously a precise quantitative agreement cannot be expected from such a simple model.

Finally, we present in Fig. 5 an $\omega/T$ scaling plot of $1/\tau$ and $m^*/m - m^*(0)/m$ for our model, as well as a direct comparison to experimental data. We emphasize that $\omega/T$ scaling does not hold exactly for either of these quantities within our Planckian model. This is due to the fact that the real part of the self-energy behaves logarithmically at low $T$ and thus leads to violations of scaling, as also clear from the need to retain a finite cutoff $\Lambda$. However, approximate $\omega/T$ scaling is obeyed to a rather high accuracy, as shown in Fig. 5a, b and discussed in more

details analytically in Supplementary Information Sec. C. Panels c and d allow for a direct comparison between the scaling properties of the theoretical model and the experimental data, including analytical expressions of the approximate scaling functions derived in Supplementary Information Sec. C. These functions stem from an approximate expression for the conductivity, Eq. (S13), that displays exact $\omega/T$ scaling. The approximation made in deriving them explains why the scaling functions differ slightly from the numerical data in Fig. 5a, b. Note the similar difference with the experimental data in Fig. 5d.

## Discussion

In this article, we have shown that our experimental optical data for LSCO at $p = 0.24$ display scaling properties as a function of $\hbar\omega/k_B T$ which are consistent with Planckian behavior corresponding to a scaling exponent $\nu = 1$. We found that the accuracy of the data scaling depends on the choice of the parameter $\epsilon_\infty$ relating the optical conductivity to the measured dielectric permittivity, and that optimal scaling is achieved for a specific range of values of this parameter.

From both a direct analysis of the optical data and by requiring optimal scaling, we demonstrated that the low-frequency optical effective mass $m^*(\omega \approx 0, T)/m$ displays a logarithmic dependence on temperature. This dependence, also a hallmark of Planckian behavior, is qualitatively consistent with that reported for the specific heat (quasiparticle effective mass)[47,48]. We showed that the coefficient of the logarithmic term can be made quantitatively consistent between these two measurements if a specific relation exists between the spectral weight $K$ and the ratio $m/m_e$ of the band mass to the bare electron mass. Interestingly, we found that a realistic tight-binding model satisfies this relation. The low-frequency optical scattering rate $1/\tau$ extracted from our scaling analysis displays a linear dependence on temperature, consistent with the $T$-linear dependence of the resistivity that we measured on the same sample, with a quite good quantitative agreement found between the $T$-linear slopes of these two measurements.

We have introduced a simple theoretical model which relies on the assumption that the single-particle inelastic scattering rate (imaginary part of the self-energy) displays $\hbar\omega/k_B T$ scaling properties with $\nu = 1$ and that its angular dependence along the Fermi surface can be neglected. These assumptions are consistent with angular dependent magnetoresistance measurements[21]. The model involves a dimensionless inelastic coupling constant $g$ as a key parameter. We calculated the optical conductivity based on this model and showed that it accounts very well for the frequency dependence (Fig. 3) and $\omega/T$ scaling properties (Fig. 5) of our experimental data.

A key finding of our analysis is that the calculated optical conductivity displays an *apparent* power-law behavior with an effective exponent $\nu^* < 1$ over an extended frequency range relevant to experiments (Figs. 3 and 4). We were able to establish that $\nu^*$ depends continuously on the inelastic coupling constant $g$ [Eq. (12) and Fig. 4a]. This apparent power law is also clear in the experimental data, especially when displaying the data for $|\sigma|$ and $\arg(\sigma)$ as a function of frequency. Hence, our analysis solves a long-standing puzzle in the field, namely the seemingly contradictory observations of Planckian behavior with $\nu = 1$ for the resistivity and optical scattering rate versus a power law $\nu^* < 1$ observed for $|\sigma|$ and $\arg(\sigma)$. We note that the apparent exponent $\nu^*$ reported in previous optical spectroscopy literature varies from one compound to another, which is consistent with our finding that $\nu^*$ depends on $g$ and is hence not universal. For our LSCO sample, the measured value of $\nu^*$ leads to the value $g \approx 0.23$.

The logarithmic temperature dependence of both the optical effective mass and the quasiparticle effective mass is directly proportional to the inelastic coupling constant $g$. We emphasize that this is profoundly different from what happens in a Fermi liquid. There, using the Kramers–Kronig relation, one sees that the effective mass enhancement (related to the low-frequency behavior of the real part of the self-energy) depends on the whole high-frequency behavior of the

imaginary part of the self-energy. In contrast, in a Planckian metal obeying $\omega/T$ scaling, the dominant $\ln T$ dependence of the mass is entirely determined by the low-energy behavior of the imaginary part of the self-energy, see Eq. (9). Based on this observation, we found that the slope of the $\ln T$ term in the effective mass and specific heat is consistent with the value $g \approx 0.23$ independently determined from the effective exponent $v^*$. Using that same value of $g$ within our simple theory leads to a value of the prefactor $A$ of the $T$-linear term in the resistivity which accounts for 60% of the experimentally measured value. Quantitative agreement would require $g \approx 0.38$, corresponding to a value of $v^* \approx 0.77$ also quite close to the experimentally observed value $v^* \approx 0.8$. It is also conceivable that electron-phonon coupling contributes to the experimental value of $A$. In view of the extreme simplification of the theoretical model for transport used in the present work, it is satisfying that overall consistency between optics, specific heat and resistivity can be achieved with comparable values of the coupling $g$.

In recent works[65,70], Planckian behavior has also been put forward as an explanation for the observed unconventional temperature dependence of the in-plane and $c$-axis Seebeck coefficient of Nd-LSCO. In these works, the same scaling form of the inelastic scattering rate than the one used here was used, modified by a particle-hole asymmetry parameter. For simplicity, this asymmetry parameter was set to zero in the present article. We have checked, as detailed in Supplementary Information Sec. F, that our results and analysis are unchanged if this asymmetry parameter is included, as is expected from the fact that optical spectroscopy measures particle-hole excitations and is thus rather insensitive to the value of the particle-hole asymmetry parameter.

Finally, we note for completeness that a power-law behavior of the optical conductivity has also been observed in other materials, including quasi one-dimensional conductors[71–74] with $v^* \sim 1.5$, and three-dimensional conductors[75–78] with $v^* \sim 0.5$. In the former case, Luttinger-liquid behavior provides an explanation for the observed power law at intermediate frequencies[71], while the interpretation of the power-law behavior for materials such as $Sr/CaRuO_3$ is complicated by a high density of low-energy interband transitions[79].

Summarizing, our results demonstrate a rather remarkable consistency between experimental observations based on optical spectroscopy, resistivity and specific heat, all being consistent with $v = 1$ Planckian behavior and $\omega/T$ scaling. We have explained the long-standing puzzle of an apparent power law of the optical spectrum over an intermediate frequency range and related the non-universal apparent exponent to the inelastic coupling constant. Looking forward, it would be valuable to extend our measurements and analysis to other cuprate compounds at doping levels close to the pseudogap quantum critical point. Our findings provide compelling evidence for the quantum critical behavior of electrons in cuprate superconductors. This raises the fundamental question of what is the nature of the associated quantum critical point, and its relation to the enigmatic pseudogap phase.

## Methods

### Sample synthesis
The $La_{1.76}Sr_{0.24}CuO_4$ ($p = 0.24$) single crystal used in the present study was grown by the travelling solvent floating zone method[80]. This sample was annealed, cut and oriented along the a-b plane and polished before measuring infrared reflectivity and resistivity.

### Infrared optical conductivity
We measured the infrared reflectivity from 2.5 meV to 0.5 eV using a Fourier-transform spectrometer with a home-built UHV optical flow cryostat and in-situ gold evaporation for calibrating the signal. In the energy range from 0.5 to 5 eV, we measured the real and imaginary parts of the dielectric function $\epsilon(\omega)$ using a home-built UHV cryostat installed in a visible-UV ellipsometer. Raw data for $\epsilon(\omega)$ are presented in

Supplementary Fig. 1. Combining the ellipsometry and reflectivity data and using the Kramers–Kronig relations between the reflectivity amplitude and phase, we obtained for each measured temperature the complex dielectric function in the range from 2.5 meV to 5 eV (see Supplementary Information Sec. B and Supplementary Fig. 1). The complex optical conductivity $\sigma(\omega)$ of low-energy transitions is directly linked to $\epsilon(\omega)$ by

$$\sigma(\omega) = i\epsilon_0 \omega \left[\epsilon_\infty - \epsilon(\omega)\right]. \tag{14}$$

In this expression, $\epsilon_\infty$ is the background relative permittivity due to high-energy transitions [see Eq. (1)]. We use international SI units, where $\epsilon_0 = 8.85 \times 10^{-5}$ kS/(cm THz). In the Gaussian CGS system, $\epsilon_0 = 1/(4\pi)$. In Scaling analysis we propose and discuss in details a procedure to estimate the value of $\epsilon_\infty$. Using the value $\epsilon_\infty = 2.76$ determined there, we display in Fig. 1a, b the real and imaginary parts of the optical conductivity. In Fig. 1a, one observes a Drude-like behavior upon cooling from 300 K, characterized by a sharpening of the Drude peak in Re $\sigma$ and a maximum in Im $\sigma$ at a frequency that decreases with decreasing $T$. For temperatures below 75 K, the Drude peak is narrower than the minimum photon energy accessible with our spectrometer, 2.5 meV, which gives the impression of a gap opening in Re $\sigma$. Yet, the superconducting transition only occurs at $T_c = 19$ K. The conductivity decreases monotonically between 0.1 and 0.4 eV, before interband transitions gradually set in.

As is common for materials with strong electronic correlations, and well documented for cuprates in particular[51,52], the optical conductivity has a richer frequency dependence than that of a simple Drude model. It is convenient however to consider a generalized Drude parametrization in terms of a frequency-dependent scattering rate $1/\tau(\omega)$ and mass enhancement $m^*(\omega)/m$ introduced in Eqs. (2) and (3):

$$\sigma(\omega) = \frac{e^2 K/(\hbar^2 d_c)}{1/\tau(\omega) - i\omega\, m^*(\omega)/m}, \tag{15}$$

so that the scattering rate and mass enhancement can be determined from the optical conductivity according to:

$$\frac{1}{\tau(\omega)} = \frac{e^2 K}{\hbar^2 d_c} \mathrm{Re}\left[\frac{1}{\sigma(\omega)}\right] \tag{16}$$

$$\frac{m^*(\omega)}{m} = -\frac{e^2 K}{\hbar^2 d_c} \mathrm{Im}\left[\frac{1}{\omega\, \sigma(\omega)}\right]. \tag{17}$$

In these expressions, $d_c = 6.605$ Å is the distance between two $CuO_2$ planes, $m$ is the band mass and $K$ is the spectral weight for a single plane. The determination of $m$ and $K$ is also discussed in Sec. II B along with that of $\epsilon_\infty$. $K$ only affects the absolute magnitude of $1/\tau$ and $m^*/m$, while the choice of $\epsilon_\infty$ has a more significant influence.

### DC transport experiment
DC resistivity was measured inside a physical property measurement system (PPMS) from Quantum Design in four-point geometry on the temperature range from 300 K to 2 K. The electric contacts were made by using silver wires of 50 μm and silver paste. To increase the contact quality, contacts were annealed at 500 °C in oxygen atmosphere for an hour in order to get a resistance of a few ohms. To obtain the resistivity $\rho(T)$ as a function of temperature in the units $\Omega$ cm from the raw sample resistance $R(T)$ in $\Omega$, the length $L$, width $W$, and thickness $t$ of the sample were measured to get a geometric factor $\alpha = W \times t/L$ knowing the relation: $\rho(T) = \alpha R(T)$. Resistivity was measured at two magnetic fields $H = 0$ T and $H = 16$ T to extract the superconducting transition temperature $T_c = 19$ K at $H = 0$ T and the normal-state resistivity down to 5 K ($H = 16$ T).

## Data availability

The experimental and theoretical data generated in this study as well as the associated codes have been deposited in the Yareta database[81].

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

## Acknowledgements

A.G. acknowledges useful discussions with Andrew J. Millis, Jernej Mravlje, and Subir Sachdev. We acknowledge support from the Swiss National Science Foundation under Division II through project No. 179157 (D.v.d.M.) and support through the JSPS KAKENHI grant 20H05304 (S.O.). The Flatiron Institute is a division of the Simons Foundation. L.T. acknowledges support from the Canadian Institute for Advanced Research (CIFAR) as a CIFAR Fellow and funding from the Institut Quantique, the Natural Sciences and Engineering Research Council of Canada (PIN:123817), the Fonds de Recherche du Québec-Nature et Technologies (FRQNT), the Canada Foundation for Innovation (CFI), and a Canada Research Chair.

## Author contributions

B.M., C.B., D.v.d.M., and A.G. conceived the project. S.K. and S.O. grew the crystal. B.M., C.W.R., A.A., L.C., L.T., and D.v.d.M. carried out the experiments. C.B. and A.G. performed the calculations. B.M., C.B., D.v.d.M., and A.G. wrote the manuscript.

## Competing interests

The authors declare no competing interests.

## Additional information

**Peer review information** *Nature Communications* thanks Jungseek Hwang, and the other, anonymous, reviewer(s) to the peer review of this work. A peer review file is available.

 

