## [Peer Review File · Nature Communications]

REVIEWER COMMENTS

Reviewer #1 (Remarks to the Author):

The manuscript titled "Planckian Behavior of Cuprate Superconductors: Reconciling the Scaling of Optical Conductivity with Resistivity and Specific Heat" by B. Michon et al. demonstrated that the power-law dependence of the optical conductivity with an exponent smaller than one is consistently connected with the T -linear dependence of the resistivity and the frequency (ω)-linear dependence of the optical scattering rate using the measured optical spectrum of LSCO near the pseudogap quantum critical point and a simple T , ω -linear Ansatz for the inelastic scattering rate. This manuscript was well-organized and well-written. It contains an interesting issue on the temperature- and frequency-dependent behaviors of charge carriers in strongly correlated electron systems, particularly copper oxide superconductors. However, there are a couple of important points that should be addressed before publication in *Nature Communications*.

1. In Fig. 1(d), the effective mass m^*/m shows ~ 1.7 at near 0.4 eV, which is quite large compared with the asymptotic value, 1.0. The authors mentioned that the high value of m^*/m at near 0.4 eV is caused by interband and/or mid-infrared transitions in LSCO. The authors need to show that it is true by either referring to appropriate literature (if there is) or using their own calculations. The interband and/or mid-infrared transitions would contribute to the optical scattering rate (\hbar/τ) as well.
2. Why were different cutoff energies (Λ) used in Fig. 2(d) and Fig. 5(d)? I wonder how it looks if they use the higher cutoff energy (0.4 eV) in Fig. 5(d).
3. In the "Theory" section, the authors started from the Planckian Ansatz with $\nu = 1.0$ and then calculated various quantities to compare with the experimental results. They also introduced a new parameter ν^* and showed that the new parameter is dependent on the inelastic coupling constant (g). How is the parameter (ν^*) related to the ν variable, which is 1.0 in the model? The relationship might be interesting.
4. The authors claimed that the Planckian model was better than the Sub-Planckian model for simulating the experimental results. But, I wonder whether the authors can completely rule out the Sub-Planckian model ($\nu < 1$). The overall results (Extended Data Fig. 9) obtained using the Sub-Planckian are not so bad compared with the experimental results; only Extended Data Fig. 9(c) does not look so good for the experimental results. Is there a better parameter set for improving the Sub-Planckian results? As shown in Extended Data Fig. 2, ν and ϵ_{∞} can be critical adjustable parameters.
5. I am just curious whether the Super-Planckian model ($\nu > 1$) is possible. It might be helpful if the authors can give a comment on it.

Reviewer #3 (Remarks to the Author):

The authors have addressed all the remarks from my previous report. Therefore, I support the publication of this paper. One minor point is that Refs. 19 and 63 are now published.

Title: Planckian Behavior of Cuprate Superconductors: Reconciling the Scaling of Optical Conductivity with Resistivity and Specific Heat

Authors: B. Michon, C. Berthod, C. W. Rischau, A. Ataei, L. Chen, S. Komiya, S. Ono, L. Taillefer, D. van der Marel, and A. Georges

Brief response to all recommendations and criticisms.

Reviewer #1: The manuscript titled “Planckian Behavior of Cuprate Superconductors: Reconciling the Scaling of Optical Conductivity with Resistivity and Specific Heat” by B. Michon et al. demonstrated that the power-law dependence of the optical conductivity with an exponent smaller than one is consistently connected with the T -linear dependence of the resistivity and the frequency (ω)-linear dependence of the optical scattering rate using the measured optical spectrum of LSCO near the pseudogap quantum critical point and a simple T , ω -linear Ansatz for the inelastic scattering rate. This manuscript was well-organized and well-written. It contains an interesting issue on the temperature- and frequency-dependent behaviors of charge carriers in strongly correlated electron systems, particularly copper oxide superconductors. However, there are a couple of important points that should be addressed before publication in *Nature Communications*. Authors: We are pleased with the referee's positive overall assessment of our manuscript.

Reviewer #1: 1. In Fig. 1(d), the effective mass m^*/m shows ~ 1.7 at near 0.4 eV, which is quite large compared with the asymptotic value, 1.0. The authors mentioned that the high value of m^*/m at near 0.4 eV is caused by interband and/or mid-infrared transitions in LSCO. The authors need to show that it is true by either referring to appropriate literature (if there is) or using their own calculations. The interband and/or mid-infrared transitions would contribute to the optical scattering rate (h/τ) as well.

Authors: The values of m^*/m and h/τ are both directly proportional to the spectral weight K used for converting the dielectric function data to conductivity data [see Eq.(14)]. The spectral weight of the Drude component is about 125 meV and the integration up to 2 eV of the total (Drude + mid-infrared) optical conductivity gives $K = 211$ meV, which is the value that we used. In the absence of other contributions to the optical conductivity, the $\omega \rightarrow \infty$ limit of the generalized Drude expression gives $m^*(\omega)/m \rightarrow 1$, whereas $m^*(\omega)/m \sim 211/125$ in the intermediate energy range between the Drude and mid-infrared component, provided that these two components are well separated. Indeed, the data displayed in Fig. 1(d) show that the ratio $m^*(\omega)/m > 1$. The fact that this quantity decreases strongly as a function of increasing energy is a consequence of the fact that the Drude and mid-infrared components of these experimental data are in fact not well separated.

Reviewer #1: 2. Why were different cutoff energies (Λ) used in Fig. 2(d) and Fig. 5(d)? I wonder how it looks if they use the higher cutoff energy (0.4 eV) in Fig. 5(d).

Authors: Since this question was posed by a previous referee, we restate our answer. As our goal is to determine unknown parameters by optimizing the scaling collapse, it is logical to include in the fitting as much as possible of the data that scale and as few as possible of the data that do not scale. Figure 1c shows that $1/\tau$ appears to scale well up to 0.4 eV; there is a weak feature around 0.22 eV corresponding to the one seen in m^*/m , but this does not break the scaling, as seen in Extended Data Fig.2. As a result, the ε_∞ determined from $1/\tau$ is not very sensitive to the cutoff. On the contrary, it is very sensitive for m^*/m , as explained in Supplementary Information Sec. B. Therefore, we have included the $1/\tau$ data between 0.22 and 0.4 eV, but not the m^*/m data. In the revised Fig. 2, we show with dotted lines the excluded m^*/m data. We also point out in Supplementary Information Sec. B that this choice is not essential: fitting all data below 0.22 eV or all data below 0.4 eV, we obtain very similar $\varepsilon_\infty = 2.91$ and 3.03, respectively, and virtually the same values of $m^*(0)$.

Reviewer #1: 3. In the “Theory” section, the authors started from the Planckian Ansatz with $v = 1.0$ and then calculated various quantities to compare with the experimental results. They also introduced a new

parameter ν^* and showed that the new parameter is dependent on the inelastic coupling constant (g). How is the parameter (ν^*) related to the ν variable, which is 1.0 in the model? The relationship might be interesting.

Authors: This relation is given by Eq.(S23) and displayed as an inset in Extended Data Fig. 8.

Reviewer #1: 4. The authors claimed that the Planckian model was better than the sub-Planckian model for simulating the experimental results. But, I wonder whether the authors can completely rule out the sub-Planckian model ($\nu < 1$). The overall results (Extended Data Fig. 9) obtained using the sub-Planckian are not so bad compared with the experimental results; only Extended Data Fig. 9(c) does not look so good for the experimental results. Is there a better parameter set for improving the sub-Planckian results? As shown in Extended Data Fig. 2, ν and ε_∞ can be critical adjustable parameters.

Authors: Indeed, from optics only, the distinction between $\nu = 1$ and $\nu < 1$ is not very sharp. The first row of Extended Data Fig.2 is, in our view, the strongest case for $\nu = 1$ based only on optical data, because in each panel ε_∞ is optimized for the best scaling. So, for $\nu = 0.8$, for instance, it is not possible to achieve a better scaling than the one shown. Now, one central point of our paper is that $\nu = 1$ is the value that gives the best consistency of optics with resistivity and specific heat. If ν is very close to one, discriminating between $\ln(1/T)$ and $1/T^{1-\nu}$ in the specific heat is arguably difficult, a little less so between $\rho \sim T$ and $\rho \sim T^\nu$ in the resistivity. However, as explained in Supplementary Information D.2, the value $\nu = 0.61$ is needed in order to explain the experimental power law with $\nu^* = 0.8$, which is not close to $\nu = 1$. Furthermore, the scaling in Extended Data Fig.2 is very poor for $\nu = 0.61$.

Reviewer #1: 5. I am just curious whether the super-Planckian model ($\nu > 1$) is possible. It might be helpful if the authors can give a comment on it.

Authors: We are not aware of a model showing super-Planckian behavior, if not for the Fermi liquid with $\nu = 2$. The Planckian limit is only rigorously valid in equilibrium, hence $\nu < 1$ is not explicitly excluded for the transport. Only for Kondo impurity models one can have $\rho(T) = \rho_0 - cT^\nu$, for example for the 2-channel Kondo model the exponent is $\nu = 1/2$.

Reviewer #3: The authors have addressed all the remarks from my previous report. Therefore, I support the publication of this paper. One minor point is that Refs. 19 and 63 are now published.

Authors: We have updated these references.

REVIEWERS' COMMENTS

Reviewer #1 (Remarks to the Author):

The authors properly addressed my comments. But the comment number 3 was about the relationship between v^* and v , not that between v_{eff} and v , which the authors provided in their manuscript. Anyway, that might be a minor thing. I recommend that the present manuscript be published in Nature Communications.

Reviewer #1 (Remarks to the Author): The authors properly addressed my comments. But the comment number 3 was about the relationship between ν^ and ν , not that between ν_{eff} and ν , which the authors provided in their manuscript. Anyway, that might be a minor thing. I recommend that the present manuscript be published in Nature Communications.*

Authors: This is merely a question of notation, that can be easily clarified: ν^* and ν_{eff} represent the same concept: both describe the power-law exponent of the conductivity. Our use of different notations was motivated by the fact that ν^* as given by Eq. (11) only applies to the Planckian case $\nu = 1$, while ν_{eff} as given by Eq. (S23) only applies to the sub-Planckian case $\nu < 1$.

To avoid misunderstanding, we have changed ν_{eff} to ν^* in Supplementary Information Sec. D, Supplementary Fig. 8, Supplementary Fig. 9, and after Eq. (S23) we added “We emphasize that this formula is not valid for $\nu = 1$. The cases $\nu = 1$ and $\nu < 1$ have different analytic structures, with the consequence that the apparent exponent in Eq. (S23) is independent of the coupling g , while in the Planckian case, Eq. (12) of the main text, it depends only on g . The function (S23) is displayed as an inset in Supplementary Fig. 8. An effective exponent of 0.8, as observed experimentally in LSCO, requires $\nu = 0.61$ (see also Supplementary Fig. 9). This in turn implies a resistivity $\rho \sim T^{0.61}$, in marked disagreement with experiments, so that a sub-Planckian interpretation is not tenable.”